# A cross-cultural study of the effect of parental bonding on the perception and response to criticism in Singapore, Italy and USA

**Michelle Jin Yee Neoh**[1], **Alessandro Carollo**[2], **Andrea Bonassi**[2], **Claudio Mulatti**[2], **Albert Lee**[1], **Gianluca Esposito**[1,2,3]*

**1** Psychology Program, School of Social Sciences, Nanyang Technological University, Singapore, Singapore, **2** Department of Psychology and Cognitive Science, University of Trento, Rovereto, Italy, **3** Lee Kong Chian School of Medicine, Nanyang Technological University, Singapore, Singapore

* gianluca.esposito@ntu.edu.sg

**Data Availability Statement:** The datasets generated during and/or analysed during the current study is available in the open access

## Abstract

Parents play a primary and crucial role in emotional socialisation processes in children where individuals learn the expression, understanding and regulation of emotions. Parenting practices and dimensions of the parent-child relationship have been associated with social and emotional processes in children. As criticism involves negative emotional reactions and emotion regulation, the parent-child relationship is likely to influence an individual's perception and response to criticism. Hence, the present study investigated the relationship of parental bonding and the perception and response to criticism in three different countries–Singapore, Italy and USA. Adult participants ($n = 444$) completed the Parental Bonding Inventory (PBI) and measures of criticism. Parental care, overprotection and country were found to be significant predictors of a tendency to perceive criticism as destructive. Higher levels of parental care predicted a lower tendency to perceive criticism as destructive while higher levels of parental overprotection predicted a higher tendency to perceive criticism as destructive. US American participants were found to have a significantly higher tendency to perceive criticism as destructive compared to Italian and Singaporean participants. The findings align with past research on the role of the parent-child relationship in the socio-emotional development of children as well as providing insight into a specific aspect in social interaction; perception and response to criticism, being affected. Future studies can look to investigate this relationship further in different countries in light of cultural variation in parenting styles and emotion experience, expression and regulation.

## Introduction

Through emotional socialisation, individuals learn to express, understand, and regulate emotion during childhood [1], and these abilities are closely linked to social interactions of children [2]. Emotions are learned within the family too, where parents play a primary role [3] and children learn about emotions and emotion regulation through their parents' responses to their emotions [4]. In a heuristic model put forth by [1], parental emotion-related socialisation

institutional data repository (DR-NTU) at the link: https://doi.org/10.21979/N9/AUNUY9.

**Funding:** This research was supported by grants from the NAP SUG to GE (M4081597, 2015-2021).

**Competing interests:** The authors have declared that no competing interests exist.

behaviours are influenced by both the characteristics of (i) the child, (ii) the parent, (iii) the culture and the aspects of the specific context in which they occur. For example, such parental emotion-related socialisation behaviours include parental reactions to children's emotional expressions, discussions regarding emotions with children and parental emotional expressiveness, which can be used to teach children and model appropriate emotional control and expression in accordance with situational demands [1, 3]. Consequently, these parental emotion-related socialisation behaviours are likely to have an effect on (i) emotional experience, (ii) emotional expression, (iii) emotion regulation and emotion-related behaviour, (iv) acquisition of regulatory processes, (v) understanding of emotions and emotion regulation, (vi) quality of the parent-child relationship and (vii) schemas about the self, relationships and the world [1]. The impact of parents on children's emotional processes are familiar from empirical studies, which found that positive, supportive parental interactions (i.e. warm, sensitive responses) to children's emotions are associated with (i) emotional competence [5], (ii) positive emotion self-awareness [6] and (iii) children's emotion regulation [7, 8]. Conversely, negative, unsupportive parental reactions (i.e. punitive or dismissive responses) are associated with (i) emotion dysregulation [7–9], (ii) socially incompetent behaviour [10] and (iii) social maladjustment [11]. Thus, parents, their parenting styles, and parent-child interactions are crucial in the emotional development and abilities of children.

Parental criticism in response to children's behaviour is an example of parental emotion-related socialisation, where criticism can be defined as negative evaluative feedback received from other people in social interactions [12, 13]. Often construed as an unpleasant experience, negative emotional reactions towards criticism are considered normative [14], in that people are inclined to feel threatened by criticism, observable not just in one or two social contexts but in numerous aspects of life [15]. In this view, criticism can be seen as a threat to the need for social belonging, or the fundamental human need that drives social bonding and the formation of attachments, interactions and relationships [16]. As a result, criticism, when construed negatively by the receiver, is a distressing experience which activates emotions and thoughts such as feeling upset or hurt, indicating its nature as a form of hurtful communication. Given that negative emotion reactions tend to be a normative response towards criticism and parents play a key role in emotional socialisation, it can be expected that differences in parenting styles would influence how individuals perceive and respond to criticism in their social interactions.

## Parental bonding and parenting practices

The quality of parenting practices and the parent-child relationship influence the formation of attachment [17]. Previous studies have shown that parental bonding is associated with mental health and adjustment [18, 19]. Parental bonding and representations of attachment bonds are closely linked to children's emotional development and emotion regulation [20]. For example, children and youths with exposure to impaired parental relationships are at higher risk of a wide range of adverse outcomes including anxiety, depression and antisocial personality, compared to those with healthy parental relationships [21–23]. The quality of the parent-child attachment in adolescents has also been suggested to have pervasive effects across the lifespan [24, 25]. In assessing the quality of parenting, parental bonding is frequently assessed with the Parental Bonding Instrument [26], which includes the factors of *care* and *overprotection*. The *care* factor measures the extent of affection and warmth in the parent-child relationship and refers to sensitivity of parents towards the child's needs. Parental care has been shown to be related to emotion regulation abilities [3]. Specifically, maternal care was found to have a negative association with the use of maladaptive emotion regulation abilities and a lack of emotional

awareness while paternal care was linked to difficulties in emotion regulation [27]. In addition, maternal care appears to be a critical factor in regulating (i) self-esteem, (ii) extent of trust, (iii) proper socialisation and (iv) emotional health. The *overprotection* factor measures the extent of controlling, overprotective behaviours exhibited by the parent which refers to excessive restrictions placed on the child such as emotional, physical and psychological restrictions. One kind of overprotective parental behaviours is psychological control which includes psychologically and emotionally manipulative parental behaviours or techniques that are not responsive to children's psychological and emotional needs [28] such as guilt induction, love withdrawal and invalidation of feelings [29] and has been associated with a negative self-concept and low self-esteem [30]. Psychological control is likely to undermine emotion regulation [31] as the emotionally manipulative nature of psychological control means that parental love and acceptance is conditional on children's behaviour. A study found that paternal psychological control has been linked to emotional symptoms in adolescents, and this relationship was found to be mediated by difficulties in emotion regulation [32]. In general, high care and low overprotection characterise ideal parenting practices. For example, individuals with overprotective fathers tend to approach others cautiously whereas those with caring fathers tend to have increased abilities in interacting with others without inhibition [33]. Similarly, high maternal overprotection levels coupled with low levels of affection have been positively correlated with feelings of being forsaken and emotional instability [33]. Hence, it can be expected that differences in parental bonding and parental practices will influence an individual's perception and response to criticism encountered in social interactions given the role of parenting in social and emotion-related processes.

## Cultural differences and criticism in relationships

There are different societal norms for experience, expression and regulation of emotion [34, 35]. For example, Japanese learn not to express negative emotions in the presence of others whereas such a tendency is not as prevalent in Americans [36]. In addition, Japanese were observed to suppress anger in close relations but express it freely toward strangers while in contrast, Americans reported feeling disgust and sadness towards ingroup members and happiness to outgroup members more so than Japanese [37, 38]. Research on cross-cultural differences tend to be grounded in the two distinct models of the self [39] and the framework of individualism and collectivism [40].

Individualistic cultures tend to promote individual needs, wishes and desires over those belonging to the group and individuals are encouraged to express themselves and their feelings, influence others and develop individuality, which is deemed to be important [39, 41]. Individualistic cultures have an independent self-construal where the basic unit of society is the individual with groups existing to promote the individual's well-being [39]. An independent self-construal is associated with a positive view of the self and a tendency to enhance the self with positive information [42–44] and a need to behave consistently with one's own attitudes and beliefs [45–47] which is compatible with values of individualism [40, 48]. On the other hand, collectivist cultures widely recognise and formalise hierarchy and status with social positions clearly defining roles and normative behaviour [49]. Collectivistic cultures have an interdependent self-construal where the group is the core unit of society in which individuals adjust to the group to maintain societal harmony [39]. The experience of the self is derived from relational attributes such as social roles, obligations and group memberships. In order to fit into the group, individuals change themselves and not influence others [41]. Western cultures–North American, Western European countries–tend to be individualistic while Eastern cultures–East Asian–tend to be collectivistic and most cross-cultural studies have compared Western versus Eastern cultures [50].

In the context of cross-cultural differences in studies on criticism, there are also different sociocultural norms and expectations for the kinds of behaviour that warrant criticism as well as the levels of criticism experienced by individuals across cultures [51] where distinct models of the self across cultures may have implications on how individuals perceive and respond to criticism. A positive view of the self and need for self-enhancement may lead to more negative emotional responses to criticism in individualistic cultures compared to collectivistic cultures. For example, studies have shown that Japanese people tend to exhibit self-critical tendencies in comparison to Europeans and/or Americans such as accepting negative self-relevant information more readily [52, 53]. In addition, US Americans responded more assertively to criticism compared to Asian counterparts (Japanese, Chinese) [54]. Cultural differences in the experiences of success and failure may also contribute to differences in the perception and response to criticism. For example, American individuals tended to select success situations as relevant to their self-esteem as compared to Japanese individuals, who tended to select failure situations [44]. A meta-analysis also found a significant cross-cultural effect on self-enhancement where Westerners showed a clear self-serving bias while East Asians did not [55]. Hence, cultural differences in terms of individualism and collectivism suggest that individuals from different cultures would show difference in the perception and response to criticism.

**Cultural differences in emotional experience and regulation.** Between different cultures, norms regarding emotional and social competence and the beliefs surrounding one's emotions and their expression [34] where what is perceived to be socially or emotionally competent behaviour varies with culture. Emotional competence refers to the use of intrapersonal and interpersonal emotional information and has been defined to be the individual differences in identifying, understanding, expressing, regulating and using one's own emotions and those of others [56, 57]. Firstly, there is cultural variability in emotional experience [58] as culture constrains how emotions are felt and expressed in different cultural contexts. Culture has a role in shaping the ways people should feel in certain situations and how emotions are expressed. Social and cultural environments all influence emotion [39, 59, 60]. A consistent finding has been that Western culture is related to high arousal emotions (e.g. happiness, enthusiasm, fear) while Eastern culture is related to low arousal emotions (e.g. contentment, misery, calm). In terms of positive emotions, the arousal level of ideal affect, defined as "affective state that people ideally want to feel" [61], differs according to culture. Due to the motivation to behave in order to feel the emotions they want to experience [61], people tend to experience the emotions considered to be ideal in their culture. For example, Americans were reported to prefer high arousal emotional states such as excitement [62] or enthusiasm [63] compared to East Asians. Similarly, studies have also found that conceptions of happiness in Americans emphasised on experiencing high arousal positive emotions such as being upbeat whereas the Chinese conception focused on being solemn and reserved [64] and the Japanese conceptualised happiness with low arousal emotional states [65].

Secondly, emotion understanding and emotion regulation are more culture-specific [66, 67]. Cultural factors are critical in informing the effectiveness of emotion regulation strategies [68] and individuals engage in and benefit when emotion regulation strategies consistent with cultural goals and self-concepts are employed [69, 70]. Findings from previous studies also showed that cultures emphasising social order and hierarchy had (i) a greater tendency for emotion suppression and (ii) positive correlations between emotion suppression and emotion reappraisal whereas cultures emphasising affective autonomy and egalitarianism had (i) a lower tendency for emotion suppression and (ii) negative correlations between emotion suppression and emotion reappraisal [69]. Specifically, individualistic cultures tend to find cognitive reappraisal more effective in managing negative affect whereas collectivistic cultures tend to display emotion suppression as it mitigates the impact of negative emotional states on others

by reducing the risk of disrupting group harmony [71]. This can be observed from (i) a positive correlation between use of emotion suppression and depressed mood scores in European American groups but not in Chinese participants [72] and (ii) correlation between the use of suppression and value placed on interpersonal harmony in Chinese participants [73]. A series of empirical findings on the use of emotion regulation strategies in different cultures support these observations. First, a study found that European Americans showed (i) an association between habitual use of suppression and self-protective goals and higher negative affect and (ii) an association between induced suppression and poor interpersonal responding and adverse perceptions of others but these associations were reduced in Asian Americans [71]. Second, a downregulation pattern of cardiovascular response to an anger provocation task was only observed in Asian Americans who valued emotional control and not in European Americans who valued emotional control [74].

In summary, culture is inextricably linked to the expression, experience and regulation of emotion and the salient differences between cultures in these various facets of emotion highlight the importance of examining how different cultures perceive and respond to criticism.

**Cultural differences in parenting styles.** As we have seen that cultural groups each embodies particular characteristics and possesses different beliefs and behaviours, these beliefs and behaviours then constitute valued competencies to be communicated to new members of the group. Essentially, it can be expected that in terms of parenting, different cultural groups possess distinct beliefs and behave in unique ways. Consequently, differences between individuals of different cultures can be largely attributed to these patterns of caregiving unique to each culture [75, 76]. In other words, culture contributes to shaping parenting and parental cognitions, that in turn shapes parenting practices [77, 78]. Cultural comparisons in parenting have documented differences across parenting goals, values, practices and parent-child interactions [79, 80]. For example, the desired parenting childrearing goals are independence, individualism, social assertiveness, confidence and competence in the dominant Western culture in the United States [76]. On the other hand, traditional Asian families emphasise interdependence, conformity, emotional self-control and humility, which is in stark contrast, or the antithesis to core values of parenting in Western cultures [81]. Taken together with the discussion of the role of parents in emotional socialisation of children, cultural variations in parenting styles and practices would also influence the development of social and emotional competence.

In the context of criticism, the presence of cultural differences in the use and expression of praise and criticism by parents reinforces the importance of investigating parenting practices across cultures on the perception and response to criticism. Western parenting is typically characterised by parental warmth, indicated by verbal and emotional expressions such as kissing and praising [82] whereas Asian parenting tend to be less likely to show outward affection and verbal expressions of love [80, 83], instead conveying affection through instrumental support, devotion, close monitoring and support for education [82]. In addition, it has been suggested that cultural differences in parental responses to children's success and failures may be responsible for how children themselves respond to success and failures, thus possibly accounting for the well documented cultural differences in response to performance. In Asian cultures, an emphasis is placed on self-improvement with effort being an integral part of this self-improvement [84] compared to Western cultures where the emphasis is placed on self-enhancement and the possession of positive attributes [42, 52]. Hence, Asian parents tend to downplay children's success and highlight failure and conversely, American parents may highlight children's success and downplay failure [85]. For example, American mothers provide positive feedback in order to fulfil their role in protecting and building children's self-esteem whereas Taiwanese mothers deem self-esteem to be unimportant and may even interfere with

a child's receptivity to correction from others [86]. This differential employment of positive feedback by parents of different cultures further highlights the role of cultural differences in parenting practices in influencing individual differences in the perception and response to criticism.

## Significance and aim of present study

The present study aims to investigate the relationship between parental bonding and the perception and response to criticism in samples from three different countries–Singapore, Italy and the United States of America (USA). Including samples from these three different countries provides insight into cultural differences in the perception and response to criticism, which is a pertinent variable to examine in light of the cultural forces underlying prevalence and employment of criticism in social interactions as pointed out earlier and parenting across various cultures as well as those shaping parenting practices.

Based on the literature on parenting practices, we expect to observe differences in the perception and response to criticism as measured by (i) sensitivity to criticism, (ii) attributions of criticism and (iii) perceived criticism. Our hypotheses are as follows:

Hypothesis 1: Parental bonding would significantly predict measures of the perception and response to criticism.

Hypothesis 2: There would be cultural differences in perception and response to criticism between the three countries (Singapore, Italy and USA).

Specifically, due to differences between individualist and collectivist cultures, it is expected that USA which has a more individualistic culture would show more negative perceptions and reactions to criticism compared to countries with more collectivistic cultures such as Singapore. While both USA and Italy are considered individualistic countries, USA has been found to be strongly individualistic and showed greater individualism compared to Italy [87, 88]. Hence, it is expected that Italy would show less negative perceptions and reactions to criticism than USA, but more than that in a collectivistic culture such as Singapore.

## Methodology

### Participants and procedure

Three samples of participants of different nationalities were recruited from three different countries; Singapore, Italy and the United States of America (USA) (Table 1). Inclusion criteria were Singaporean, Italian and American citizens respectively aged 18–35 years old. The study was approved by the Institutional Review Board (IRB) at the Nanyang Technological University (IRB Number: 2019-10-037) for recruitment of online participants of other nationalities and written informed consent was obtained from participants before completing the questionnaire.

Participants from the Singaporean sample were recruited ($n$ = 150, male = 75, female = 75) through a psychology undergraduate course and compensated with course credits. Participants from the Italian sample were recruited ($n$ = 84, male = 42, female = 42) through a psychology undergraduate course and compensated with course credits. Participants from the American sample were recruited from Amazon Mechanical Turk ($n$ = 210, male = 103, female = 107) and received monetary compensation. All participants answered questions regarding their demographic information and completed the following scales on a questionnaire hosted on Qualtrics.

**Table 1. Descriptive statistics for age, gender and relationship status.**

| | Country | | |
|---|---|---|---|
| | **Singapore** | **Italy** | **USA** |
| | **(n = 150)** | **(n = 84)** | **(n = 210)** |
| **Gender** | | | |
| **Male** | 75 | 42 | 103 |
| **Female** | 75 | 42 | 107 |
| **Age** | Mean = 22.69 | Mean = 21.64 | Mean = 24.12 |
| | (SD = 1.81) | (SD = 1.91) | (SD = 2.16) |
| **Relationship status** | | | |
| **Currently in a romantic relationship (%)** | 27.3 | 45.2 | 63.8 |
| **Have previously been in a romantic relationship (%)** | 24.7 | 40.5 | 25.7 |
| **Never been in a romantic relationship (%)** | 48 | 14.3 | 10.5 |

## Questionnaire measures

**Sensitivity to Criticism Scale.** The Sensitivity to Criticism scale (SCS) [89] is a 30-item, self-report measure that assesses two elements of sensitivity: (i) perceptual–extent to which a situation is *perceived* as criticism and (ii) emotional–degree of emotional response. The SCS presents participants with hypothetical situations in which participants may perceived criticism directed at themselves. Participants then rated the extent they would (i) view the situations as criticism and (be) hurt by the situation on a 7-point Likert scale. A shortened version of the scale with 15 items was used in the present study where participants rated their perceptions and emotional response in the situations with the criticism originating from one's romantic partner, friend, parents or a stranger. The SCS has been found to be internally consistent with a Cronbach's alpha of 0.92–0.94 across three studies and a test-retest reliability after a 6-month interval of $r = 0.82$ reported during the construction of the scale [89].

**Attributions of Criticism Scale.** The Attributions of Criticism Scale (ACS) [90] is a 22-item questionnaire assessing the attributions–positive and negative–individuals make about the intentions underlying their romantic partner or relative's criticism. Participants rated their attributions on a 5-point Likert scale with regard to a romantic partner, friend, parents and a stranger. The ACS has been found to demonstrate a two-factor structure corresponding to positive and negative attributions [91], good test-retest reliability and convergent validity with perceived criticism measures.

**Perceived criticism measure.** Perceived criticism (PC) [92] ratings were obtained from each participant for romantic partners, friends, parents and strangers. PC was assessed with the question "How critical is (the relative/a stranger) which was rated on a 10-point scale [92]. This item has been described as the gold-standard measure of perceived criticism. PC ratings have also demonstrated high predictive validity, correlated with expressed emotion [92, 93] and high test-retest reliability [92]. The measure also includes an item "When your relative/a stranger criticizes, how upset do you get?" which was also rated on a 10-point scale. This item has been shown to have predictive validity in predicting poor clinical outcomes for patients with bipolar disorder, obsessive-compulsive and panic disorder with agoraphobia [94, 95].

**Parental Bonding Inventory.** The Parental Bonding Inventory (PBI) [26] measures an individual's perceived parent-child attachment characterized in terms of two dimensions; *care* and *overprotection* or *control*. *Care* relates to the extent to which affection and sensitive parenting were perceived from both parents while *overprotection* looks at the extent to which individuals perceive their parents to be (i) implementing excessive control and/or (ii) impeding their

growth towards independence. The PBI is a 25-item form completed for the individual's mother and father. The PBI has been shown to have a high test-retest reliability after a period of 20 years and was relatively unaffected by changes in mood [96, 97].

## Analytic plan

First, principal components analysis was conducted on the measures of perception and response to criticism: (i) sensitivity to criticism, (ii) attributions of criticism (positive and negative) and (iii) perceived criticism. Using the derived principal components, stepwise regression for each principal component was conducted by inserting the following predictors into (i) PBI Care (Mother), (ii) PBI Care (Father), (iii) PBI Overprotection (Mother), (iv) PBI Overprotection (Father), (v) Relationship type of the source (of criticism), (vi) Country of origin, (vii) Gender.

## Results

### Principal components analysis

Scores of the different measures of criticism; (i) SCS, (ii) ACS Positive, (iii) ACS Negative and (iv) PC were submitted to a principal components analysis. Three principal components accounting for a total of 90.06% of the variance were used for the regression analysis. The principal components are summarised in Table 2. Figs 1 and 2 show the biplots of the principal components and how the criticism measure scores are loaded. Based on how the criticism measure scores loaded onto the principal components, it is proposed that the principal components can be conceptualised to represent: (i) PC1 –a tendency to perceive criticism as destructive, (ii) PC2 –a tendency to perceive criticism as positive and (iii) PC3 –general levels of criticism perceived in an individual's social environment.

### Regression analysis

A stepwise regression was conducted to determine which of the following variables; (i) PBI Care (Mother), (ii) PBI Care (Father), (iii) PBI Overprotection (Mother), (iv) PBI Overprotection (Father), (v) Relationship type of the source (of criticism), (vi) Country of origin, (vii) Gender, were significant predictors of the three principal components. A model was derived for each principal component. The $p$-value used for entry into the model is 0.05 and the $p$-value for removal is 0.1. The final model output is shown in Tables 3–5.

**Parental bonding.** With regards to a tendency to perceive criticism as destructive (PC1), all of the parental bonding measures were found to be significant predictors by order of entry as follows: (i) maternal care ($t = -7.965$, $p < .001$), (ii) paternal overprotection ($t = 6.120$, $p < .001$), (iii) maternal overprotection ($t = 3.992$, $p < .001$) and (iv) paternal care ($t = -2.208$, $p = .027$). These results suggest that (i) as parental care increases, the tendency to perceive criticism as destructive decreases whereas (ii) as parental overprotection increases, the tendency to perceive criticism as destructive increases.

**Table 2. Table of principal components.**

|  | PC1 | PC2 | PC3 |
|---|---|---|---|
| **Standard deviation** | 1.27 | 1.10 | 0.88 |
| **Proportion of variance** | 40.51% | 30.40% | 19.15% |
| **Cumulative proportion** | 40.51% | 70.91% | 90.06% |

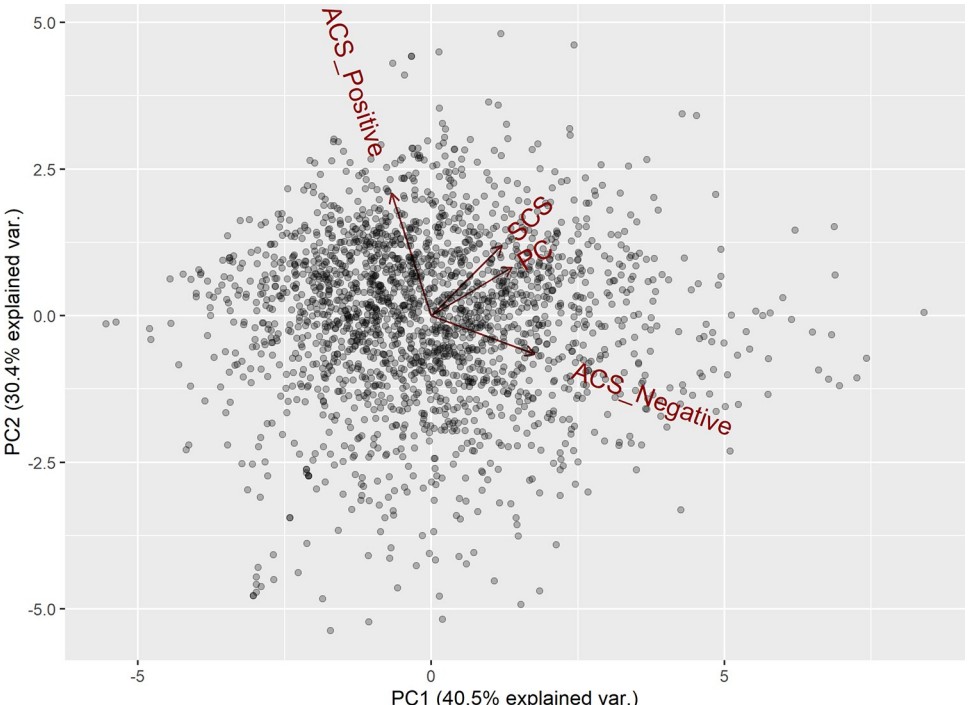

**Fig 1. Biplot of principal component 1 and principal component 2.**

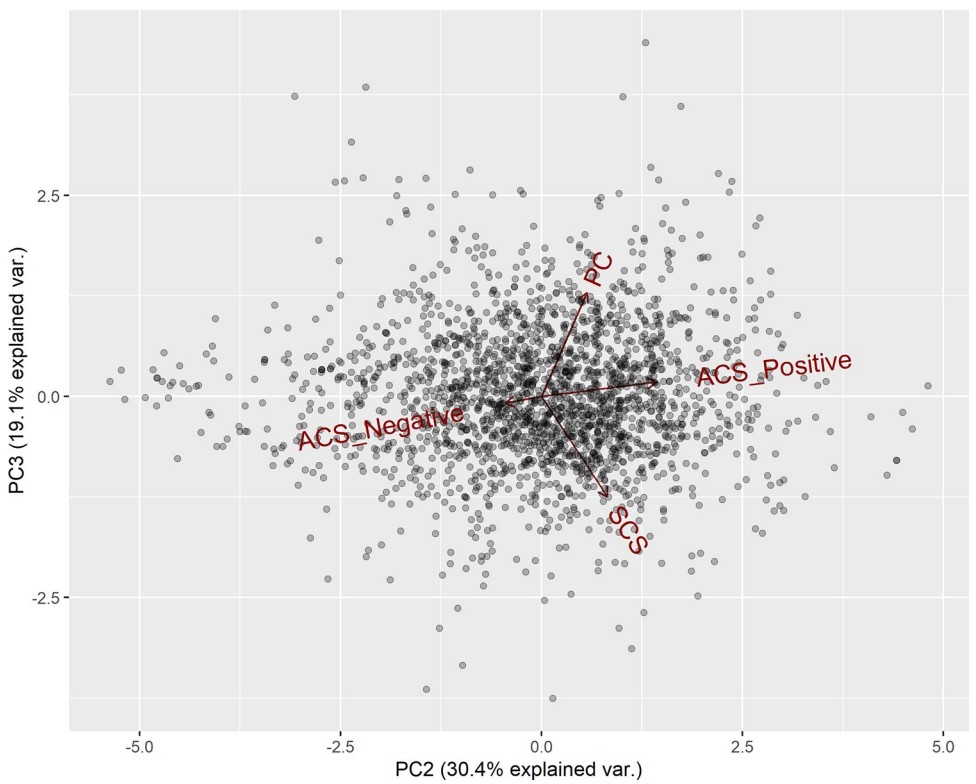

**Fig 2. Biplot of principal component 2 and principal component 3.**

**Table 3. Table of final model in stepwise regression for PC1.**

| Principal Component | Step | Variables | $R^2$ | β | | t-value | p-value |
|---|---|---|---|---|---|---|---|
| PC1 –Tendency to perceive criticism as destructive | 1 | PBI Care (Mother) | 0.091 | -0.030 | | -7.965 | < .001 |
| | 2 | Source | 0.178 | Mother | 0.335 | 4.376 | < .001 |
| | | | | Father | 0.398 | 5.202 | < .001 |
| | | | | Romantic partner | 0.379 | 4.588 | < .001 |
| | | | | Stranger | 1.124 | 14.681 | < .001 |
| | 3 | PBI Overprotection (Father) | 0.218 | 0.025 | | 6.120 | < .001 |
| | 4 | Country | 0.226 | Singapore | 0.058 | 0.815 | 0.415 |
| | | | | USA | 0.270 | 3.953 | < .001 |
| | 5 | PBI Overprotection (Mother) | 0.232 | 0.016 | | 3.992 | < .001 |
| | 6 | PBI Care (Father) | 0.233 | -0.007 | | -2.208 | 0.027 |
| | | $R^2 = 0.233$ (F = 61.7, $p < .001$), Adjusted $R^2 = 0.230$ | | | | | |

With regards to a tendency to perceive criticism as constructive (PC2), the parental bonding measures found to be significant predictors by order of entry as follows: (i) paternal care ($t = 4.860, p < .001$), (ii) paternal overprotection ($t = 3.979, p < .001$) and (iii) maternal care ($t = 3.500, p < .001$). These results suggest that (i) as parental care and paternal overprotection increases, the tendency to perceive criticism as constructive increases.

With regards to general levels of criticism perceived in an individual's social environment (PC3), only maternal care was found to be a significant predictor ($t = -4.885, p < .001$). These results suggest that as maternal care increases, the higher the general levels of criticism one perceives in his/her social environment.

**Cultural differences.** Country was found to be a significant predictor of (i) PC1 (Singapore; $t = 0.815, p = .415$, USA; $t = 3.953, p < .001$) and PC3 (Singapore; $t = 1.171, p = .242$, USA; $t = -8.006, p < .001$). These results suggest that (i) there is a higher tendency to perceive criticism as destructive in USA compared to Singapore and Italy and (ii) a lower general level of criticism perceived in one's social environment in the USA compared to Singapore and Italy. The means for the three principal components by country can be found in Table 6.

## Discussion

### Parental bonding

All parental bonding measures were found to be significant predictors of a tendency to perceive criticism as destructive. Firstly, the tendency to perceive criticism as destructive decreases with higher levels of maternal and paternal care, suggesting that warmth in the parent-child

**Table 4. Table of final model in stepwise regression for PC2.**

| Principal Component | Step | Variables (by order of entry) | $R^2$ (cumulative) | β | | t-value | p-value |
|---|---|---|---|---|---|---|---|
| PC2 –Tendency to perceive criticism as constructive | 1 | Source | 0.324 | Mother | 0.356 | 5.838 | < .001 |
| | | | | Father | 0.098 | 1.602 | 0.109 |
| | | | | Romantic partner | 0.302 | 4.607 | < .001 |
| | | | | Stranger | -1.318 | -21.641 | < .001 |
| | 2 | PBI Care (Father) | 0.334 | 0.012 | | 4.860 | < .001 |
| | 3 | PBI Overprotection (Father) | 0.338 | 0.012 | | 3.979 | < .001 |
| | 4 | PBI Care (Mother) | 0.342 | 0.010 | | 3.500 | < .001 |
| | | $R^2 = 0.342$ (F = 150.4, $p < .001$), Adjusted $R^2 = 0.339$ | | | | | |

**Table 5. Table of final model in stepwise regression for PC3.**

| Principal Component | Step | Variables (by order of entry) | $R^2$ | β | | t-value | p-value |
|---|---|---|---|---|---|---|---|
| **PC3 –General levels of criticism perceived in an individual's social environment** | 1 | Country | 0.057 | Singapore | 0.061 | 1.171 | 0.242 |
| | | | | USA | -0.390 | -8.006 | < .001 |
| | 2 | Source | 0.104 | Mother | 0.259 | 4.648 | < .001 |
| | | | | Father | 0.265 | 4.762 | 0.109 |
| | | | | Romantic partner | -0.191 | -3.189 | .001 |
| | | | | Stranger | -0.142 | -2.547 | 0.011 |
| | 3 | Gender (Male) | 0.122 | 0.245 | | 6.790 | < .001 |
| | 4 | PBI Care (Mother) | 0.133 | -0.011 | | -4.885 | < .001 |
| | | $R^2 = 0.133$ (F = 38.8, $p < .001$), Adjusted $R^2 = 0.230$ | | | | | |

relationship contribute to how individuals perceive and approach criticism. Drawing on the idea that individuals develop mental representations of their relationships with others and that negative attributions have been correlated with greater perceived destructive criticism [91, 98], it is possible that individuals who grow up with a parent-child relationship characterised by warmth and positive affect would be less likely to make negative attributions about a parent's criticism, and subsequently the criticism encountered in other social relationships. On the other hand, negative parenting such as negative affect predicts emotion dysregulation [99, 100], which could affect their perception of criticism received. Hence, individuals who experience greater parental care in their relationships with their parents could be less likely to think of others as putting them down or expressing disapproval as well as experiencing a lower extent of the negative emotional reaction towards the comments they receive. Secondly, the tendency to perceive criticism as destructive increases as parental overprotection increases. A possible interpretation of this finding is that controlling behaviours such as psychological control could influence how individuals perceive criticism due to its role in undermining emotion regulation [31]. Due to the emotionally manipulative nature of psychological control where parental love and acceptance is conditional on children's behaviour, psychologically controlling parents could lead to the creation of a coercive, unpredictable or negative emotional climate of the family, serving as one of the possible avenues through which the family context influences emotion regulation of children [3, 101]. This environment exerts pressure on children to conform to parental authority, resulting in children's emotional insecurity and dependence. Psychological and emotional manipulation such as guilt induction, love withdrawal, invalidation of feelings [29] has also been associated with a negative self-concept and low self-esteem [30]. It is possible that individuals who grew up in such a family environment with a parent-child relationship characterised by psychological control would be both more likely to make negative attributions about a parent's criticism, such as the intention of the parent to control or withhold love when expectations are not met, and also to perceive parents as being critical. In addition, difficulties with emotion regulation and a low self-esteem resulting from

**Table 6. Table of means for principal components by country.**

| | Singapore | Italy | USA |
|---|---|---|---|
| **PC1 –Tendency to perceive criticism as destructive** | -0.168 | -0.307 | 0.234 |
| **PC2 –Tendency to perceive criticism as constructive** | -0.002 | -0.047 | -0.017 |
| **PC3 –General levels of criticism perceived in an individual's social environment** | 0.245 | 0.147 | -0.204 |

psychological control by parents could lead to individuals experiencing a greater negative emotional response towards comments made by parents. Consequently, the mental representations these individuals have of their parent-child relationships may also influence their view of their other social relationships. In addition, difficulties with emotion regulation in couples has also been found to predict more hostile perceived criticism [102], which further supports the idea that parent-child relationships characterised by low warmth and high psychological control could lead to increased tendencies to perceive criticism as destructive due to poorer emotion regulation abilities. Lastly, it is of note that findings in the present study of the role of parental bonding in the perception and response to criticism are in line with growing literature that find love-related behaviours and influence of fathers as being equally, even significantly, more influential than that of mothers in mood related disorders such as the development of depression and other psychological problems [103] as well as behaviour problems [104].

On the other hand, the source of criticism accounted for most of the variance in the tendency to perceive criticism as constructive compared to parental bonding measures, which were small but significant predictors, in the final model. Firstly, this finding dovetails with the process model of constructive criticism proposed by [105] where one of the main aspects for feedback to be perceived as constructive criticism was that it needed to originate from a respect-worthy source of criticism and be embedded in perceptions of care. Secondly, another possible explanation for the present finding could be that comments construed as constructive criticism are less ambiguous in terms of their definition and content. According to the process model of constructive criticism, the properties for a message to be perceived as constructive criticism include being well-intentioned, appropriate targeting and provide guidance for improvement [105], which appear to suggest a specific profile for comments to be construed as constructive criticism. Another study also similarly found an overwhelming consensus across an undergraduate sample in perceptions of constructive criticism where definitions nearly always included an element of improvement, which was noted to suggest an understanding of constructive criticism messages as intending to improve performance [106]. Hence, the present finding, along with those from previous studies, suggest that the tendency to perceive criticism as constructive may be more contingent on characteristics of the message itself and the relational context due to the mostly unambiguous nature of messages of constructive criticism.

## Cultural differences

There were differences between countries in terms of the tendency to perceive criticism as destructive where US Americans showed a higher tendency to perceive criticism as destructive. A possible explanation for this finding is the difference in communicative styles between US Americans compared to Singaporeans and Italians, in terms of the differences in communicative styles between individualistic and collectivistic cultures [107–110]. A *low-context* communication styles characterised by assertiveness and valuing talk tend to be used by US Americans. In contrast, *high-context* communication styles characterised as non-assertive, where less value is placed on talk and greater reliance is placed on the ability to intuit what is being implied, needed or wanted, tend to be used by Japanese and Chinese. In addition, a previous study showed that US Americans were more likely to engage in active criticism, where the preferred forms of criticism were found to be "through constructive suggestions", "in a direct way", "sarcastic remarks", "angrily" and "in an insulting way" [111]. On the other hand, the Japanese sample in this study were more likely to engage in passive criticism, where they more frequently reported that they would "attempt not to show dissatisfaction", "express dissatisfaction to a third person" and express such dissatisfaction "nonverbally", "ambiguously"

and "humourously" [111]. Notably, given that criticism in Americans is usually expressed with sarcasm, anger and insultingly, it can be expected that there would be a higher tendency to perceive criticism as destructive as opposed to the indirect nature of the expression of criticism by the Japanese. Hence, differences in communication styles could have a role in how criticism is expressed in different cultures and consequently, differences in expression of criticism could possibly explain differences in the tendency to perceive criticism as destructive across cultures.

Another possible explanation for the observed difference between countries in tendency to perceive criticism as destructive is a difference in ingroup/outgroup distinction between cultures. The ingroup/outgroup distinction has been found to be particularly important in collectivistic cultures where members of collectivistic cultures exhibit differentiation according to group membership more sharply than members of individualistic cultures [112] and group membership is stronger and more permanent for members of collectivistic cultures [113]. A significant body of work has looked at the intergroup sensitivity effect (ISE) [114] in group-directed criticism, which refers to the finding that criticism from ingroup members is generally received more positively than criticism from outgroup members, even when the content of the criticism is identical. The cause of the ISE has been posited to be mediated by an attributional bias, where ingroup members are attributed to have more constructive motives than outgroup critics in accordance with existing literature indicating that ingroup members tend to be trusted more than outgroup members [115, 116] and more favourable outcomes are expected from ingroup compared to outgroup members [117]. When applied to the context of criticism, group membership can be used as a heuristic to distinguish individuals with constructive motives, which would then subsequently impact the responsiveness towards the delivered message. As such, members of collectivistic cultures with a stronger ingroup/outgroup distinction would be more likely to demonstrate ingroup bias and trust towards ingroup members, resulting in more constructive motives being attributed to criticism received. On the other hand, members of individualistic cultures would be less likely to attribute constructive motives towards criticism from others and be more likely to attribute destructive motives towards criticism instead. Hence, this could possibly explain a higher tendency to perceive criticism as destructive observed in the USA sample, which has an individualistic culture compared to the Singapore sample, which has a collectivistic culture. Although this work investigated criticism at the group level, it is reasonable to expect that similar mechanisms of ingroup/outgroup distinctions on attributions ascribed to the source of criticism to individual-directed criticism are at play, especially since attributions have been shown to be related to perceptions of destructive and constructive criticism [91, 98].

Cultural differences were also found in the general levels of criticism perceived in an individual's social environment where lower levels of criticism were perceived in the USA sample compared to the Singapore and Italy samples. This finding is largely consistent with the idea that there are cultural differences in the expression of emotion and affection in terms of the use of praise by parents although the data in the present study is not well placed to support this claim and future studies can look into investigating the perception of praise and criticism as expressions of affection in different cultures.

## Implications

The findings in the present study largely supported the hypotheses in that (i) parental bonding has an effect and (ii) cultural differences were present in the perception and response to criticism although both of these were not observed in all of the principal components derived in the present study.

Firstly, the significance of parental bonding measures in predicting the tendency to perceive criticism as destructive reinforce the importance of the parent-child relationship in emotional socialisation and social information processing, specifically, in shaping perceptions and responses to criticism, given the empirical associations of perceptions of destructive criticism with relationship quality and mental health outcomes.

Secondly, cultural differences in the tendency to perceive criticism as destructive highlights the importance of being aware of how feedback is delivered and subsequently, construed by individuals from different cultures, especially in educational and organisational settings where students or employees stand to learn and improve one's thinking or task performance. Negative feedback was found to evoke defensiveness, anger and repudiation of feedback in an organisational setting [118]. Perception of feedback as destructive criticism can also possibly lead to feelings of anger or tension and lower goals and self-efficacy [119] where goals were adjusted downward after receiving negative feedback in contrast to an upward adjustment after receiving positive feedback [120].

Thirdly, findings from the present study could also be of note for clinicians in the treatment of patients with mental health disorders that have been shown to relate to levels of criticism–such as depression, schizophrenia and eating disorders. Clinicians could take note of possible cultural differences in how patients construe feedback received from clinicians, which could possibly affect patient engagement and treatment outcomes.

## Limitations and future directions

While we have investigated samples from three different countries in the present study, it is important to note that even within the broad categories of the commonly employed individualistic versus collectivistic framework, individual countries classified under either category are still characterised by idiosyncrasies that make them distinct from one another. For example, intra-regions of Asia should be investigated for their distinctness given that Asian cultures tend to nurture distinct conceptions of individuality incurring from the fundamental relatedness between people [39]. More specifically, a study found significant differences in measures of both intrapersonal and interpersonal emotional competence between Asian countries–namely, Myanmar, Japan, China and Bangladesh [121]. Furthermore, acculturation of parenting practices in migrant populations is another area that can be studied. In migrant families, both parents and children are exposed to values of the host culture of the country they have migrated to and their original culture continues to influence family values and norms and new behaviours that are developed in the context of the country they have migrated to [122]. Hence, the present study can be replicated in more countries and comparisons can also be made between migrant populations to local populations in order to gain a more holistic perspective of the nature of criticism in social interactions across different cultures.

## Conclusion

The importance of the parent-child relationship and parenting practices in emotional socialisation and socio-emotional development of children has been consistently established and emphasised and present findings on the significance of parental bonding in predicting the perception and response to criticism support this empirical association. In this respect, the parent-child relationship is embedded as part of a wider context of the family unit and patterns of interactions within different familial relationships can be expected to have an influence on the nature and quality of these relationships.

## Acknowledgments

All participants are gratefully acknowledged. We would like to acknowledge members of the Social Affective Neuroscience Lab at NTU for their assistance in the completion of this project.

## Author Contributions

**Conceptualization:** Michelle Jin Yee Neoh, Gianluca Esposito.

**Data curation:** Michelle Jin Yee Neoh.

**Formal analysis:** Michelle Jin Yee Neoh, Gianluca Esposito.

**Funding acquisition:** Gianluca Esposito.

**Investigation:** Michelle Jin Yee Neoh, Alessandro Carollo, Andrea Bonassi.

**Methodology:** Michelle Jin Yee Neoh, Alessandro Carollo, Andrea Bonassi, Gianluca Esposito.

**Visualization:** Michelle Jin Yee Neoh, Gianluca Esposito.

**Writing – original draft:** Michelle Jin Yee Neoh, Claudio Mulatti, Albert Lee, Gianluca Esposito.

**Writing – review & editing:** Michelle Jin Yee Neoh, Claudio Mulatti, Albert Lee, Gianluca Esposito.

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
